# Emerging Role of Neutrophil Extracellular Traps in Gastrointestinal Tumors: A Narrative Review

**DOI:** 10.3390/ijms24010334

**Published:** 2022-12-25

**Authors:** Yujun Zhang, Jingjing Song, Yiwei Zhang, Ting Li, Jie Peng, Haonan Zhou, Zhen Zong

**Affiliations:** 1Department of Gastrointestinal Surgery, The Second Affiliated Hospital of Nanchang University, 1 MinDe Road, Nanchang 330006, China; 2HuanKui Academy, Nanchang University, Nanchang 330006, China; 3Nanchang University School of Ophthalmology & Optometry, Jiangxi Medical College, Nanchang University, Nanchang 330006, China; 4Queen Marry College, Nanchang University, Nanchang 330006, China; 5The Second Clinical Medical College, Nanchang University, Nanchang 330006, China

**Keywords:** neutrophil extracellular traps, gastrointestinal neoplasms, molecular targeted therapy, biomarkers, quantitation

## Abstract

Neutrophil extracellular traps (NETs) are extracellular fibrous networks consisting of depolymerized chromatin DNA skeletons with a variety of antimicrobial proteins. They are secreted by activated neutrophils and play key roles in host defense and immune responses. Gastrointestinal (GI) malignancies are globally known for their high mortality and morbidity. Increasing research suggests that NETs contribute to the progression and metastasis of digestive tract tumors, among them gastric, colon, liver, and pancreatic cancers. This article explores the formation of NETs and reviews the role that NETs play in the gastrointestinal oncologic microenvironment, tumor proliferation and metastasis, tumor-related thrombosis, and surgical stress. At the same time, we analyze the qualitative and quantitative detection methods of NETs in recent years and found that NETs are specific markers of coronavirus disease 2019 (COVID-19). Then, we explore the possibility of NET inhibitors for the treatment of digestive tract tumor diseases to provide a new, efficient, and safe solution for the future therapy of gastrointestinal tumors.

## 1. Introduction

Gastrointestinal tumors are a general term for benign and malignant tumors that originate in the digestive tract. Gastrointestinal neoplasms are mainly composed of tumors occurring in the colon, rectum, stomach, pancreas, and liver, causing a heavy burden on public health care. The above-mentioned tumors in the gastrointestinal tract are extremely lethal and deserve full attention. Colorectal cancer (CRC) of the colon and rectum represents the second most prevalent cause of cancer-related deaths, while gastric cancer (GC) ranks as the fourth leading contributor to death worldwide according to cancer statistics [1]. Recently, the research on the management of gastric cancers has been relatively abundant, with new advances in surgical resection plus lymph node dissection, chemotherapy, targeted therapy, etc. Patients with gastric cancer have improved overall survival compared to previous rates. However, gastric cancer is still responsible for more than one million new cases in 2020 and an estimated 769,000 deaths worldwide (equivalent to one in every 13 deaths globally). Pancreatic ductal adenocarcinoma (PDAC) is among the deadliest tumor types, considering that 460,000 people died in 2020 and its high level of mortality of more than 90%. PDAC is widely considered to be a typical aggressive carcinoma, and its overall 5-year survival rate continues to be low, at 6% [1]. Liver cancer ranks as the third leading cause of cancer-associated deaths and the sixth highest frequently diagnosed carcinoma worldwide. In addition, hepatocellular carcinoma (HCC) constitutes 70% to 85% of liver cancers. Despite the use of modern surgical techniques and advanced combination administrations of adjuvant and neoadjuvant chemoradiotherapy, a considerable number of patients develop distant metastases and resistance to the therapy [2]. As mentioned above, GI tumors are diverse and serious, and there is no effective therapy available to successfully control and treat this kind of disease. Therefore, new therapeutic strategies are urgently needed to improve the treatment of GI malignancies.

There is now a growing body of research demonstrating that the immune system has a part to play in tumor suppression as well as treatment. Among them, the role of neutrophils is also becoming a research hotspot in the comprehensive treatment of GI cancers [3]. NETs, composed of genomic DNA modified with proteins, play a key role as neutrophil weapons against GI cancer progression [4,5]. Preliminary studies have shown that neutrophils can also capture and kill pathogens through NETs when stimulated by pathogens or phorbol myristate acetate (PMA) [6]. In recent years, national and international scholars have proved the contribution of NETs to the pathogenesis of sepsis, thrombosis, autoimmune diseases [7,8], and COVID-19 [9,10]. Inflammatory bowel disease (IBD) [11], ulcerative colitis (UC), and Crohn’s disease (CD) [12] are characterized by chronic unresolved intestinal mucosal inflammation involving innate and adaptive immune responses. Neutrophils are key players in the inflammatory milieu of the intestinal mucosa in IBD, perpetuating and enhancing intestinal inflammation by forming NETs [13,14]. Studies have shown more vital associations between NETs and cancer, including promoting angiogenesis [15] and tumor cell migration [16], inducing a hypercoagulable state [17], and even stimulating dormant cancer cells [18]. Therefore, more detailed research is urgently required to elucidate the possible mechanisms underlying the association between NETs and gastrointestinal cancers. 

This review first elaborates on the formation mechanism of NETs in detail. Then, the relationship between NETs and the tumor microenvironment was investigated. According to the latest research, NETs’ effects on digestive tract tumors are introduced in detail, including tumor growth, metastasis, angiogenesis, etc. In addition, we also list the current targeted drugs that can inhibit NET production to prevent digestive tract tumor progression and discuss the development process of these drugs in clinical trials. In addition, the influence of NETs after surgical stress on GI cancers is comprehensively analyzed. Then, the different detection and quantification methods of NETs in GI tumors are briefly summarized. Finally, we highlight the correlation between COVID-19-induced NETs and cancer progression. We will also analyze the use of NET-associated inhibitor drugs in GI cancers.

## 2. NETosis

Some research has found that some neutrophils can promote tumor invasion and metastasis [19]. Subsequent studies suggested that this dual role might be attributed to two distinct subtypes of tumor-associated neutrophils (TAN): anti-tumor neutrophils (N1) and pro-tumor neutrophils cells (N2) [20]. Interferon β (IFN-β) induces neutrophils to polarize into an anti-tumor N1 phenotype, while growth factor β (TGF-β) promotes production of protumor N2 neutrophils. N1 with anti-tumor effects can induce cytotoxicity by releasing high levels of superoxide and hydrogen peroxide, destroying tumor cells and inhibiting tumor growth [21]. In contrast, N2 neutrophils can release a large number of immunoregulatory factors, such as arginase and matrix metalloproteinase 9 (MMP-9) to promote tumor development [22,23]. Interestingly, the N1 phenotype can turn into the N2 phenotype as the tumor progresses [24]. However, these distinct neutrophil phenotypes remain undefined in humans. The effect of NETs in cancer pathogenesis may have a similar background and be influenced by the number of neutrophils involved, since both N1 and N2 neutrophils can form NETs [25].

In 1996, maximal activation of neutrophils using PMA was observed, with neutrophil lobulated nuclei decondensing and aggregating, cell and nuclear membrane rupturing, and nuclear material filling the cytoplasmic space [26]. This process is distinct from apoptosis or necrosis. While these observations were previously thought to be a bizarre, unexplained mode of neutrophil death, a key advance has been the realization that this process is not just neutrophil death but reveals mechanisms of bacterial sequestration. Brinkmann first discovered a network structure induced by neutrophils and named it NETs in 2004 [27]. NETs are special ultrastructures composed of smooth fibers with diameters of 15–17 nm after killing invading bacteria. Subsequent studies found that when viruses [28], parasites [29], and fungi [30] invade our body, neutrophils form NETs by embedding various proteins outside the cell, thereby releasing depolymerized DNA. The main components combined with DNA are neutrophil elastase (NE), myeloperoxidase (MPO), histones, cathepsin G (CG), and possibly as many as 30 other molecules [31,32].

Recent studies suggest that NETs are generated by neutrophils in two ways [33,34] (Figure 1). The first way is through NADPH oxidase (NOX)-dependent lytic NETs formation. When neutrophils are stimulated by factors such as phorbol myristate acetate (PMA), NADPH oxidase is activated through the RAF/MEK/ERK or the protein kinase C (PKC) signaling pathway, leading to the synthesis of reactive oxygen species (ROS) and the activation of protein arginine deaminase 4 (PAD4). PAD4 induces the formation of citrullinated histone H3 (CitH3) in the nucleus, which in turn promotes the depolymerization of chromatin and its release by exocytosis to form NETs [35]. IL-8 is an important cytokine that is released during tumor development. It has been reported that IL-8 interacts with CXC chemokine receptor 2 (CXCR2) on neutrophils, thereby triggering the formation of lytic NETs by activating the src–p38–ERK signaling pathway [36]. It is an oxidation-dependent pathway. Notably, there are also many pathways for NETs formation that are oxidant-independent. For example, signal inhibitory receptor 1 (SIRL1) on leukocytes can be involved with particular signaling routes for NETs formation without affecting oxidant production [37]. In addition, MPO translocates to the nucleus with the help of NE [38] and promotes further chromatin depolymerization [39]. The nuclear envelope is decomposed, and the depolymerized chromatin gets released into the cytoplasm, modified by granules and cytoplasmic proteins, and then liberated from the cell. The second way is through NADPH-oxidase (NOX)-independent non-lytic NETs formation. The formation of such NETs is related to calcium influx and the generation of mitochondrial ROS independent of NOX activation [40]. Under the stimulation of *Staphylococcus aureus s* (*S. aureus*), *Escherichia coli* (*E. coli*), damage-associated molecular patterns (DAMPs), platelets, and Toll-like receptor 2/4 (TLR2/4), lymphocyte function-associated antigen 1 (LFA1) of neutrophils bind to these allergens, promoting the non-lytic NETs formation. This formation is also accompanied by chromatin de-concentration, followed by protein-modified chromatin expulsion through vesicles without disrupting the plasma membrane. The integrity of the nuclear and cellular membranes of neutrophils is not compromised under this NET formation pathway. After releasing the NETs, neutrophils are still alive and possess phagocytic and chemotactic capabilities [41]. However, relatively little research has been done on the formation of non-lytic NETs. The process of lytic NETs takes 2–4 h, whereas the process of non-lytic NETs takes 5–60 min. Interestingly, Leppkes et al. observed the release of DNA by neutrophils with alkaline cytoplasmic pH under the influence of sodium bicarbonate and resealing of the plasma membrane following the expulsion of depolymerized chromatin, which was associated with the liberation of the whole nucleus [42]. This finding may correlate features of lytic and non-lytic methods of NET formation.

## 3. NETs in the Tumor Microenvironment

The inflammatory response associated with immune cells infiltrating the tumor microenvironment (TME) is thought as an indispensable mediator leading to tumor progression and metastasis along with immunotherapy response [43,44,45,46]. Neutrophils are the predominant circulating leukocytes in humans and are often regarded as a central player in host responses to pathogens. As related studies have progressed, the proportion of neutrophils in the immune infiltration of a variety of tumors, including HCC [47], PDAC [48], CRC [49], and GC [50], has become increasingly evident. They can kill pathogens quickly through phagocytosis, but also by releasing their potent antimicrobial arsenal without phagocytosis, including oxidants [51], granular enzymes, and proteins, as well as NETs [52,53]. The existence of NETs has been identified in the TME of a wide range of solid tumors including GI cancers [54]. G-CSF release into the bloodstream has been reported to contribute to tumor recruitment of neutrophils to form NETs [55]. Furthermore, exocytosis by carcinoma cells, including GI carcinoma cells, triggers IL-8 generation and stimulates NETosis by neutrophils [56,57]. However, there is relatively little straightforward assessment of collaterals for NETs and other immune cells in the TME. NETs can inhibit the suppressive effect of immunity in the surrounding microenvironment on tumor cells by acting as a physical barrier and as a chemical that suppresses immune cells. It has become increasingly clear that the tumor-associated neutrophil (TAN) N1 type releases immunostimulatory or pro-inflammatory cytokines, comprising CXCL10, CCL3, and tumor necrosis factor (TNF)-α, which promote CD8+ T cell recruitment and activation. TGF-β in the tumor microenvironment invokes tumor-promoting N2-TAN and blocks the TGF-β-induced tumor suppressor N1-TAN [58]. TAN N1 has also been reported to produce NETs to surround malignant tumor cells, thereby protecting them from CD8+ T cell- and NK cell-mediated cytotoxicity [59]. Time-lapse confocal microscopy has shown that NETs block contact between cancer cells and cytotoxic immune cells in subcutaneous cancer in a mouse model. Furthermore, the effect of NETs may go beyond the physical barrier of cancer cells and may involve deleterious effects of mediators derived from neutrophils on NK cells and/or CTLs. However, N2-TAN is more common in the TME. Similarly to tumor-associated macrophages (TAM), N2 TAN can act by producing pro-tumor factors and impacting other cells of the immune system by immunosuppressive means, such as by triggering T cell tolerance [60]. Therefore, tumor growth and metastasis [61,62] can be suppressed and the level of immunosuppression in the TME can be reduced by depleting N2-polarized TANs, thereby increasing the activity of CD8 cytotoxic T-lymphocytes (CTLs) [20]. 

NETs can promote tumor growth by triggering the exhaustion of T cells. De Andrea et al. found a negative association between CD8+ T-cell density and the NET area in tissue microarrays of bladder and NSCLC cancer after adding CD8 dye to the multiplex panel [63]. Furthermore, a study conducted by Kaltenmeier revealed that a NET-rich environment induces and accelerates the growth of metastatic tumors and promotes an exhausted phenotype in T cells. Furthermore, it demonstrated that NETs can suppress T-cell responses via being metabolized and functionally depleted [64]. During hepatic ischemia-reperfusion injury (IRI), neutrophils can facilitate the recruitment and activation of CD8+ T cells via different cytokines and chemokines expressed in the chromatin of activated neutrophils [65]. NETs have been found to contain the immunomodulatory protein programmed death ligand 1 (PD-L1), which binds to PD-1 on activated T cells, rendering them inactive and depleted. These new findings may have clinical significance to overcome T-cell depletion and tumor progression by aiming at neutrophils and NETs [64,66].

NETs and cancer-associated fibroblasts (CAFs) promote each other and work together to support the occurrence of cancer. CAFs are not only the major component of the tumor microenvironment but also an important factor in GI tumor metastasis. They may maintain tumorigenesis by modulating the immune response to some extent [67,68]. 

Colin Hutton used single-cell high-throughput flow cytometry to classify pancreatic fibroblasts as two separate but formerly unrecognized subpopulations according to CD105 expression: CD105-negative and CD105-positive. CD105-negative fibroblasts are correlated with enhanced immune cell infiltration and immune responses against tumors, whereas CD105-positive pancreatic fibroblasts allowed tumor growth in vivo. There may be a more complex transcriptional program behind the expression of CD105 to select the direction that mediates the transformation of fibroblasts into tumor-promoting or tumor-suppressing [69]. In addition, a recent study revealed that NETs promoted the differentiation and function of lung fibroblasts stimulated by fibrosis-related medicine [70]. Based on this research, Shin Takesue et al. investigated the role of DNase I in the recruitment of activated CAFs by constructing spontaneous PDAC mouse models. The results showed that Dnase I plays a vital role in fibrotic stroma and liver metastasis [16]. The contribution of NETs in liver micrometastasis in PDAC by activating CAFs is suggested in their results. Recent studies suggest that CAF may engage neutrophils and trigger NET formation. One potential mechanism is that CAF induces the production of amyloid-β (a peptide associated with inflammatory and neurodegenerative diseases [71] and significantly upregulated in PDAC [72]), resulting in the generation of NETs in a ROS-dependent manner [73]. In turn, NET formation promotes CAF contraction, expansion, and deposition of stromal components that support tumorigenesis. Based on observations in mice, and in human PDAC and melanoma, CAFs are detected alongside NETs and showed increased expression of amyloid-β and β-secretase, which correlates with poor prognosis [73]. 

In summary, NETs promote tumorigenesis by inhibiting tumor suppressor factors like the suppressive effect of immunity and promoting tumor-promoting factors like CAFs in the tumor microenvironment.

## 4. NETs in Digestive Tract Tumors

### 4.1. Gastric Cancer

There is a certain relationship between NETs and gastric cancer. The formation of NETs was first identified in the microenvironment of GC tissues by Yiyin Zhang et al. NETs levels and neutrophil buildup was reduced in peripheral blood (PB) from neoplastic to paraneoplastic tissues. Higher levels of peripheral blood NETs at baseline are related to poorer progression-free survival (PFS). Their study also showed that NETs have better tumor serodiagnostic capabilities than routine biomarkers, like carbohydrate antigen 19-9 (CA19-9) and carcinoembryonic antigen (CEA) in GC. These suggest that NETs have a significant role in oncogenesis and tumor growth in GC [74].

NETs may promote tumor development by inducing epithelial–mesenchymal transition (EMT) and promoting vascular remodeling. It has been observed that NETs can impair endothelial cells so that trapped neoplastic cells may disseminate and form neo-metastases following adhesion to activated endothelial cells [75]. Specifically, NETs may promote tumor spread and metastasis in two ways. The first way was revealed that NETs could promote GC cell metastasis via EMT. EMT is a critical stage in the pathogenesis of GC related to spreading and metastasis [76]. This study also demonstrated that NET formation was significantly upregulated in GC patients, and its increased levels were consistent with the increased tumor stage [77]. The detailed process might be that GC enables neutrophils to induce NETosis, and then NETs were deposited in GC tissue or adhered to epithelial cancer cells, specifically without interfering with cell proliferation and the cell cycle. Another way NETs stimulate tumor initiation and progression is by affecting the expression of the ANGPT2 gene in endothelial cells. Shifeng Yang et al. found that NETs promoted ANGPT2 overexpression in human umbilical vein endothelial cells. ANGPT2 is a member of the angiopoietin (Ang) family [78], which modulates vascular remodeling and tumor growth in many pathologic situations via its differential influence on TIE2 signaling. In this study, they found that ANGPT2 overexpression in GC is not only related to poor prognosis, but may also regulates multiple biological functions. Targeting neutrophils/NETs and ANGPT2 may be a future novel approach to antitumor therapy [79].

Furthermore, NETs may affect tumor development by regulating RNA expression. Numerous studies have revealed that RNA has emerged as a direct mechanism for the transformation of healthy cells into tumor cells, which has a critical function in cancer diagnosis and prognosis [80,81], especially in GC. For example, miR-96-5p has been revealed to promote GC cell growth by direct inhibition of FOXO3 expression [82]. A mechanism study revealed that NORAD overexpression could facilitate GC cell growth by modulating the miR-608/FOXO6 pathway [83]. A recent study comprehensively analyzed the genes that are differentially expressed in GC cells treated with NETs and verified the clinical implications of NEAT1-related signaling. RNA interference of NEAT1 was found to inhibit NETs-induced AGS cell invasion, suppress miR-3158-5p expression, and enhance RAB3B expression [84]. RAB3B was positively associated with tumor staging, and miR-3158-5p was negatively associated with tumor staging. However, more basic and clinically intensive experiments to uncover the modulation of NETs by ceRNA networks in GC are urgently needed. Apart from direct contact with tumor cells, NETs could be engaged in abnormal clotting in patients with gastric cancer. NETs levels have been observed to markedly correlate with D-dimer and thrombin–antithrombin complex levels in GC patients, indicating that NETs may be involved in facilitating blood coagulation in GC progression [85]. Venous thromboembolism (VTE) is a common complication among GC patients [86,87] and is associated with high mortality [88,89]. An increase in activated platelets may lead to increased thrombus formation [90]. A recent study found that NETs induced a hypercoagulable state of platelets by upregulating the expression of P-selectin and phosphatidylserine on the cells. Compared with control mice in the inferior vena cava stenosis model, massive accumulation of NETs in tumor mouse thrombi greatly enhanced the ability to form thrombi. 

There are currently a series of corresponding treatments for the etiology of abnormal blood coagulation in gastric cancer patients caused by NETs. Of note, the combined use of sivelestat, deoxyribonuclease I, and activated protein C significantly abolished the procoagulant activity (PCA) of NETs [91]. Abnormal blood coagulation in cancer can be treated not only by inhibiting the procoagulant activity of NETs but also by reducing the production of NETs. Salvia miltiorrhiza (Danshen) is a plant used medicinally for the treatment of cancer [92], and its root extract can prevent neutrophils from transferring to metastatic sites by inhibiting the activities of MPO and NADPH oxidase (NOX), thereby obstructing the formation of NETs [93]. This was also validated by the latest research that showed that one of the bioactive polyphenols extracted from Danshen called SAA, can attenuate oxidative stress, inflammation, and neutrophil NETosis to ameliorate acute lung injury [94]. Compared with traditional anticancer drugs, tumor nanotherapeutics have various unique advantages. A recent study used Abraxane/human neutrophil (NE)-cell-based neoadjuvant chemotherapy with radiation therapy for efficacious cancer management. This cellular drug-based approach to adjuvant radiotherapy increased the release of inflammatory factors that led to NE homing to the tumor site while destroying the tumor. Explosive release of Abraxane triggers remarkable cancer suppression after excessive activation of Abraxane/NEs to generate NETs at the neoplastic regions [95]. 

All in all, NETs can directly contact tumor cells, regulate gene expression to promote tumor development, and may also be involved in abnormal blood coagulation in patients with gastric cancer, leading to aggravation of symptoms. The multiple action pathways of NETs also provide directions and means for corresponding drug development and treatment.

### 4.2. Colorectal Cancer

A positive feedback effect between CRC (colorectal cancer) progression and NET formation has been identified and several mechanisms have been proposed to trigger the formation of NETs in the CRC microenvironment. For example, in KRAS-mutated CRC cells, potential prognostic and predictive markers for CRC [96,97] have been shown to accelerate the deterioration of CRC by promoting NET production through exosomal activation of neutrophils [97]. Furthermore, tumor cell-driven IL-8 expression can recruit and activate neutrophils to generate more NETs, thereby aggravating CRC progression [57]. The link between KRAS and IL-8 is that CRC cells may transfer mutated KRAS to neutrophils via exosomes, thereby promoting NETs formation by modifying IL-8 and ultimately leading to CRC exacerbation. Another study suggested that poly P expressed by CD68+ mast cells in CRC was one of the factors that triggered the release of NETs from neutrophils. Therefore, the detection of CD68+ poly P-expressing mast cells may also represent another promising prognostic marker like KRAS-mutated CRC cells in CRC [98].

Advanced CRC is prone to distant metastasis, the most common site of which is the liver [99]. NETs contribute to the metastasis of colorectal cancer. Numerous studies have shown that approximately 25–30% of CRC patients will develop coetaneous liver metastasis, most of which show a marked increase in NET formation [100,101]. In a mouse model of hepatic IR injury, it was found that increased levels of postoperative NETs promote the progression of liver metastasis and were associated with decreased disease-free survival in a cohort of patients with colorectal liver metastases who underwent curative hepatectomy [102]. IL-8 overexpression activated neutrophils to form NETs, forming a positive loop that boosts CRC liver metastasis [103]. NETs induce colorectal cancer liver metastasis by interacting with some molecules that promote colorectal cancer metastasis. Amino acids 21–25 at the extracellular N-terminus of CCDC25 (a transmembrane protein expressed in CRC cells) were found as the binding sites for NET-DNA by Yang et al. in their remarkable study aimed to explain why CRC cells have the propensity to spread to the liver. Liver metastasis of CRC cells can be induced by initiating the β-Parvin–RACI–CDC42 pathway to enhance cancer cell motility [104]. Another newly identified molecule that promotes metastasis in colorectal cancer is carcinoembryonic antigen-related cell adhesion molecule 1 (CEACAM1), a cell adhesion molecule expressed on endothelial cells. Blocking CEACAM1 on NETs, or knocking it out in mouse models, resulting in a more than 50% reduction in colon cancer cell adhesion and migration [105]. In addition, using high-resolution stimulated emission depletion (STED) microscopy, Antonia M et al. found that citrullinated NETs were significantly associated with high histopathological tumor grades and lymph node metastasis. Their findings suggested that NETs activate an epithelial–mesenchymal transition-like process in CRC cells and may play a crucial role in the metastatic progression of CRC [106]. 

After realizing the possible role of NETs in colorectal cancer metastasis, some studies further verified this relationship and explored new directions for future treatment by inhibiting the formation of NETs. Using preclinical murine models of lung and colon cancer combined with in vivo video microscopy, Rayes et al. found that NETs functionally contributed to metastatic progression and that blocking NETosis through multiple measures markedly inhibited spontaneous metastasis to the lung and liver [100]. DNase I can disrupt NET formation by cleaving DNA strands. Xia et al. used an adeno-associated virus (AAV) gene therapy vector which specifically expressed DNase I in the liver. NET formation was significantly inhibited in tumor tissues with AAV-DNase I treatment. This result demonstrated that AAV-mediated DNase I liver gene transfer can be a potential therapeutic target for preventing CRC metastasis [101].

It is well known that cancer-related thrombosis is strongly associated with poor prognosis, and patients with CRC are generally at higher risk for venous thrombosis. NETs, as intermediate substances, also mediate the mutual promotion of cancer and blood coagulation. A study revealed a novel link between coagulation, neutrophilia, and complement activation, in which NETs are also important factors. It found that elevated circulating lipopolysaccharide (LPS) induced the upregulation of complement C3a receptor on neutrophils and activation of the complement cascade, which led to NETosis, induction of N2 polarization, and coagulation, thus accelerating tumorigenesis. This provides a favorable explanation for the promotion of tumor development through coagulation [107]. On the other hand, the occurrence of cancer cells can also further promote blood coagulation. Cancer cells may promote NET formation through TLR9 and mitochondria-activated protein kinase signaling in LPS-injected CRC mice [108]. The generation of NETs in cancer patients resulted in a marked increase in thrombin–antithrombin (TAT) complexes and fibrin fibers [109]. This eventually led to a shortened neutrophil clotting time in colorectal cancer patients compared with healthy controls. Knowing the relationship between NETs and thrombus in cancer patients, we can use NETs and substances related to them as biomarkers. For instance, Brice et al. first showed that circulating DNA (cirDNA) might appear as a marker of NETs and speculated that a significant portion of the cirDNA amount is derived from NETs in metastatic colorectal cancer (mCRC) patients. High levels of antiphospholipid antibodies (aPL) in the plasma may trigger thrombosis in cancer patients [110]. Furthermore, a study showed for the first time that aPL was closely associated with NET markers and cirDNA in cancer patients, suggesting that examining these markers may help prevent blood clots in cancer patients [111].

### 4.3. Liver Cancer

The two key immune cells of the liver that contribute to the capture of pathogens are neutrophils and Kupffer cells [112]. Although Kupffer cells play a key role in antibacterial defense, in some cases Kupffer cells after capturing bacteria can also become potential pathogens [113]. However, NETs released by neutrophils can prevent the escape of infected Kupffer cells. Since neutrophils lack CRIg receptors, they cannot directly capture bacteria in the blood, but they can directly capture and kill bacteria by releasing NETs. While NETs can kill bacteria, excessive formation of NETs can lead to worsening of inflammation. Several factors are associated with the development of HCC, including hepatitis B or C virus, metabolic disorders, nonalcoholic steatohepatitis, or alcohol intoxication, with some of these etiologies are potentially associated with NETs [114]. NETs have been found to contribute to the pathogenesis of autoimmune liver disease, non-alcoholic steatohepatitis (NASH), and other chronic liver diseases [115], which are closely related to the occurrence of HCC. It suggests that NETs may play a significant role in HCC development.

NASH is a progressive, inflammatory fatty liver disorder and one of the risk factors for HCC. The chronic inflammation caused by steatosis through production of NETs plays an important role in its pathogenesis. Studies have found that elevated free fatty acids such as linoleic acid and palmitic acid in NASH can stimulate liver-infiltrating neutrophils to form NETs, which then cause monocyte-macrophage infiltration and increases in inflammatory cytokines such as IL-6 and TNF-α. Increased inflammatory cytokines affect the tumorigenic, inflammatory microenvironment of the liver and participate in the occurrence and development of HCC. Furthermore, a study has shown that the production of inflammatory cytokines is caused by an increase in NET-activated TLR4 signaling. The active TLR4 signaling induced naive T cells to differentiate into Tregs. The differentiation of naive T cells into Tregs led to the malignant transformation of epithelial cells and the occurrence of liver cancer [116]. That is to say, fatty liver can lead to the formation of NETs, affect the inflammatory microenvironment, and promote the occurrence of liver cancer. Consistent with the findings of this study, inhibition of NETs by DNase treatment or PAD4 gene knockout did not affect the progression of fatty liver but may slow the growth of HCC [117]. Along the same line, NETs enriched with IL-1β and IL-17A were found to participate in the hepatic inflammatory process of patients with NASH [118]. Patients with NASH are at high risk for venous thromboembolism and high morbidity and mortality. Therefore, a deeper study of the neutrophil/NETs/IL-1β/IL-17α axis in both upstream and downstream of polarized Th17 cell-driven adaptive immune responses is warranted. 

NETs can not only affect the local inflammatory microenvironment in NASH patients, but may also lead to the hypercoagulable state of blood in patients. A recent study found that plasma levels of NET markers were considerably higher in NASH patients than in healthy controls. NETs exert cytotoxic effects on endothelial cells, rendering them with a pro-coagulant and pro-inflammatory phenotype [119]. Subsequently, Armando also found that the release of NETs in non-alcoholic fatty liver disease (NAFLD) patients may enhance the hypercoagulant state of NASH patients [120]. These discoveries led to the recognition that NETs may have a critical role in the procoagulant events of NASH. According to the above studies, we realized the influence of NETs on the inflammation and blood coagulation of NASH patients and that NETs may induce hepatocellular carcinoma, which provides an optimal treatment strategy to reduce inflammation and prevent hepatocellular carcinoma in NASH patients or reduce the pathological hypercoagulation state of NASH patients.

Many studies have illustrated that neutrophils and the NETs they produce play complex roles in the pathophysiology of alcoholic liver disease [121,122]. Excessive alcohol consumption can stimulate neutrophils and promote the production of NETs leading to pathological damage and even liver cancer. Recently, Yang Liu analyzed potential molecular mechanisms that might engage the migration of bacterial metabolites from the intestine to the liver and the activation of NETs [123]. They found that NETs formed after LPS activation of TLR4 following chronic alcohol consumption caused elevated alcoholic steatosis, which consequently led to HCC. Hepatitis virus infection is widely known as a common cause of liver cancer. The pathogenic mechanism of hepatitis virus infection is related to NETs. A study revealed that the hepatitis B virus (HBV) can suppress NETs release by modulating the production of ROS and autophagy to evade the immune system and promote the establishment of chronic infections [124]. In addition to causing chronic infection, HBV may also promote the growth and metastasis of HCC cells. According to recent findings published in *Cancer Letters*, HBV used NETs to exacerbate the progression of hepatocellular carcinoma. Specifically, HBV induced S100A9 to activate the RAGE/TLR4–ROS signaling pathway to allow massive NET formation [125]. In addition, based on the mechanism of HBV aggravating hepatocellular carcinoma metastasis, the prediction of HBV-associated extrahepatic metastasis of HCC can use NETs as a biomarker in the future. In chronic viral hepatitis, the infection is promoted by inhibiting the function of NETs, while in fulminant viral hepatitis (FVH), NETs may aggravate liver damage. FVH is a life-threatening disease, but its pathogenesis is not fully understood. The study by Li et al. proposed a mechanism whereby in FVH, NETs exacerbate FVH liver injury by promoting fibrin deposition and inflammation. They illustrated that the formation of NETs was regulated by the fibrinogen-like protein 2 (Fgl2)–mucolipin 3–autophagy axis [126]. Due to the remarkable heterogeneity of human viral hepatitis syndromes and pathogens, the applicability of their study to human disease is unclear. We look forward to more nuanced studies to clarify the emerging role of NETs in FVH.

NETs are closely related to the development of HCC. Van der Windt et al. used neonatal streptozotocin and a high-fat diet to induce a NASH–HCC mouse model [117]. At 20 weeks, a large number of cancer cells accumulated on the liver surface of all male mice. After inhibiting the formation of NETs with drugs or knocking out the PAD4 gene, the size and number of cancer cells on the liver surface were significantly reduced. This preliminarily suggested that the expression of NETs played a role in promoting the development of HCC. Furthermore, NETs were found to promote the development of HCC by affecting the inflammation and invasion ability of liver cancer, and the pathways affected by NETs were further studied. Yang et al. found that, compared with healthy people, serum NET marker MPO-DNA levels were increased in HCC patients, and it was more significant in patients with metastatic liver cancer. Through in vitro research, they confirmed that the increased expression level of NETs can enhance the cytotoxicity and invasiveness of liver cancer cells. NETs induced the inflammatory response of HCC cells by up-regulating COX2, thereby activating the TLR4/9 signaling pathway, and enhancing the metastasis ability of liver cancer cells. This result suggests that NETs can promote tumor inflammation and liver cancer metastasis [127]. In addition, subsequent research illustrated that NETs could also downregulate tight junction molecules on adjacent endothelial cells, thereby facilitating tumor infiltration and metastasis [128]. In addition, Xiangqian found that NETs-associated cathepsin G (cG) facilitated HCC metastasis in vitro as well as in vivo. Clinically, the expression of cG protein in tumor tissues was closely related to the prognosis of HCC patients [129]. From the above three molecular signals and signaling pathways related to NETs, we can see that the generation of NETs is accompanied by the regulation of various signaling molecules and signaling pathways, which jointly affect the occurrence and development of HCC. Furthermore, a clinical retrospective investigation found that higher NET levels in preoperative sera were associated with shorter relapse-free survival/overall survival in HCC patients [130]. These results suggest a role for NETs as a prognostic factor in patients with liver malignancies.

The fundamental impact of mitochondrial metabolism on all steps of tumorigenesis (i.e., malignant transformation, tumor progression, and response to therapy) is widely recognized [131]. The role of NETs in the relationship between mitochondria and tumors has also been preliminarily recognized. NETs can enhance the function of mitochondria, which can then promote cancer cell growth. CRC research has found that NETs can directly stimulate Hepa1-6 and Huh 7 cell proliferation by enhancing mitochondrial function and biogenesis in vitro [132]. Neutrophil elastase (NE) released by NETs activated TLR-4 on cancer cells, inducing PGC-1α upregulation, increased mitochondrial biogenesis, and accelerated growth. The interaction between mitochondria and NETs is bidirectional, and mitochondria can also affect the state of cancer cells by affecting the formation of NETs. A recent study found that neutrophils in HCC patients contained high levels of mitochondrial ROS and formed NETs enriched with oxidized mtDNA in a mitochondrion-adherent manner. Hepatocellular carcinoma can stimulate NETs rich in oxidized mtDNA, which play a crucial role in promoting inflammation and metastasis [133]. Related to this mechanism, a new finding demonstrated that mesenchymal stromal cell-derived extracellular vesicles (MSC-EVs) have a nanotherapeutic effect [134]. They can inhibit the formation of local NETs by transferring functional mitochondria to intrahepatic neutrophils and repairing their mitochondrial function. The restricted synthesis of NETs can further prevent and limit the inflammation and metastasis of HCC. That is to say, after recognizing the relationship between NETs and mitochondria in promoting hepatocellular carcinogenesis, we can use NETs and overexpressed mitochondria or even the mediators of their interaction as therapeutic targets.

Notably, the NP-neutrophil targeting approach can be used to prevent NET formation in GI cancers [135]. Some recent studies, which revealed the relationship between nanoparticles (NPs) and liver injury [136], used nanoparticles to adjust the level of NETs to prevent and treat liver inflammation. A study demonstrated that zinc oxide nanoparticles (ZnO-NPs) could elevate the levels of malondialdehyde (MDA), decrease superoxide dismutase (SOD) levels, and increase the levels of NETs in the liver of crucian carp [137]. In addition, DNase I can prevent ZnO–NPs-induced liver injury, which provides new insights into the immunotoxicity of ZnO–NPs in fish. DNase I is a key enzyme in the degradation of NETs induced by NPs and the alleviation of liver inflammation also was found in the human liver. Some studies also used nanoparticles (NPs) to further reveal the relationship between NETs and liver inflammation. Qianru Chi et al. have revealed the mechanism underlying NETs formation in polystyrene nanoparticle (PSNP) exposure-induced liver inflammation [138] providing an applicable and efficient target for anti-cancer therapies.

Although the use of nanoparticles and other strategies combined with NETs provides a useful direction for clinical anticancer treatment, it has not yet been put into clinical practice, and cancer surgery or liver transplantation is still relatively common. In these surgical operations for liver cancer, hepatic ischemia-reperfusion injury (IRI) often occurs [139]. Recent studies have implicated NETs in the pathogenesis of I/R. Injury-associated molecular patterns such as histones and HMGB1 protein are released from damaged hepatocytes and induce NETs formation by activating neutrophil TLR4 and TLR9 [140]. Using a mouse model of hepatic IRI, it was found that induction of NETs can further activate platelets, leading to systemic immune thrombosis and distal organ damage [141,142]. There are corresponding strategies for this. Researchers found that the level of NETs in mice was reduced and liver damage was improved by pretreating mice undergoing I/R with allopurinol and N-acetylcysteine, which aimed to reduce circulating superoxide [143]. These results suggest that antioxidant therapy may prevent liver I/R by attenuating NETs formation. In addition, there are some biomolecules and related signaling pathways that can be used to inhibit the formation of NETs to suppress the degree of IRI damage. A recent study found that human thrombomodulin (rTM) significantly inhibited the formation of NETs by neutrophils through blocking Toll-like receptor 4 and the downstream extracellular signal-regulated kinase/c-Jun NH2 terminal kinase and nicotinamide adenine dinucleotide phosphate (NADPH)/reactive oxygen species/peptidyl arginine deiminase 4 signaling pathways. This effect contributed to the reduction in hepatocyte apoptosis, alleviation of rat liver IRI, and improvement of liver function [144].

### 4.4. Pancreatic Cancer

Pancreatic cancer is often associated with poor prognosis, in which neutrophils are abnormally recruited into the tumor microenvironment, leading to tumorigenesis [145]. Significantly increased NET formation and decreased NETs degradation were observed in the serum of PDAC patients [146]. The etiology of PDAC patients may be related to abnormally high NETs in the body. More specifically, the amount of NETs is negatively associated with recurrence-free survival and OS rate and could serve as a separate factor to evaluate the prognosis of PDAC patients [147]. A study found that the mechanism of the role of NETs in pancreatic cancer is related to the overexpression of the tissue inhibitor metalloproteinase-1 (TIMP1), whereas abrogation of NETs formation or TIMP1 expression was associated with prolonged survival [148]. This effect depended on the interaction of TIMP-1 with its receptor CD63 and subsequent induction of MEK/ERK signaling. Notably, we can use signaling molecules in this pathway leading to pancreatic cancer as prognostic targets. Plasma levels of TIMP-1 and NETs combined with the clinically established marker CA19-9 have a better prognostic value than CA19–9 alone as TIMP1 overexpression and NET formation are inseparably associated with PDAC progression [149]. This pioneering study is very important because TIMP1 has been proposed as a potential serum marker to detect early stages of familial PDAC [150], which allows for more precise prognoses. In addition, lysine(K)-specific demethylase 6A (KDM6A) is a frequently mutated tumor suppressor gene in PDAC. A recent study analyzed the effects of KDM6A loss on the immune microenvironment of PDAC tumors. They demonstrated that KDM6A-deficient PDAC cells alter the immune microenvironment by increasing CXCL1 secretion and neutrophil recruitment. Loss of KDM6A was associated with increased tumor-associated neutrophils and NET formation. Their study provides evidence for targeting the CXCL1–CXCR2 signal axis in low-KDM6A tumors [151].

Recent research showed that NETs play an emerging role in PDAC progression and inflammation-associated metastasis. Just as NETs promote the proliferation of cells such as gastric and colorectal cancers, NETs also boost the proliferation of pancreatic cancer cells. Recently, it was found that NET-associated IL-1β was involved in the EMT process of pancreatic cancer through the EGFR–ERK pathway [152]. NETs can also facilitate tumor proliferation by activating pancreatic stellate cells that form dense fibrous stroma by interacting with RAGE receptors [153]. In addition, the formation of NETs favors the metastasis of tumor cells. A study found the generation of NETs around metastatic tumors. The same study further demonstrated that NETs induce EMT of tumor cells, thereby promoting their migration and invasion. Thrombomodulin can attenuate the malignancy of cancer cells and prevent pancreatic cancer metastasis to the liver by degrading HMGB1 derived from NETs [154]. This further confirms the role of NETs in tumor cell metastasis. NETs can also serve as intermediate substances in the signaling pathway of other biomolecules acting on PDACs. Numerous studies have demonstrated that DDR1 whose expression in PDAC negatively correlates with clinical outcomes [155], may play a key role in collagen-driven tumorigenesis in PDAC [156,157]. Jenying Deng et al. established that collagen-induced CXCL5 production is mediated through the DDR1/PKCθ/SYK/NF-κB signaling pathway. CXCL5 promoted cancer cell invasion and metastasis by inducing TAN to form NETs [158]. Citrullinated histone arginine deiminase 4 (PAD4) is critical in NET release [159]. Treatment with the PAD4 inhibitor GSK484 reduced NET formation and completely inhibited pancreatic tumor growth in a xenograft mouse model [73]. 

In addition to the above-mentioned key substances affecting the growth of pancreatic tumors, NETs can also be used as factors affecting blood coagulation in patients with pancreatic cancer. In a clinical trial, in patients undergoing pancreatectomy for pancreatic cancer, the formation of circulating NETs in pancreatic tumor tissue was significantly reduced, although intraoperative lidocaine infusion which can decrease bleeding combined with epinephrine did not improve overall or disease-free survival [160]. In a clinical study, NET formation appeared to correlate with procoagulant activity [161]. NET release converted endothelial cells to a procoagulant phenotype, resulting in increased levels of thrombin, factor Xa, and fibrin formation and rapid clotting associated with healthy control samples. Therefore, NETs can aggravate the condition by affecting tumor proliferation, migration, and coagulation in PDAC patients. NETs and neutrophils have already been verified to contribute to the enhancement of venous thrombosis (VTE) in animal experiments in mice [162]. Importantly, in the above experiments, neutrophil depletion, or DNase I administration reduced thrombus size in tumor-bearing mice but not in control mice. However, not all venous thromboembolism in cancer patients is associated with the formation of NETs. Maraucher et al. showed that H3cit histone modification, an established marker of NETs formation, was not associated with other types of cancer, but only correlated to venous thromboembolism in patients with certain types of cancer including lung cancer and PDAC [163]. The latest study described a novel hypercoagulable state mechanism of thrombosis including NETs released by activated neutrophils and the critical role of extracellular vesicles (EVs) involved in platelet and coagulation activation [164]. As for the treatment of hypercoagulable states, one study found that chloroquine inhibited NET formation and the associated hypercoagulable state in an orthotopic mouse model of PDAC and samples from patients with PDAC [165].

## 5. NETs in Surgical Stress

Despite the advances in therapeutic strategies in GI cancers, surgical resection of malignant tumors remains the mainstay treatment. Subsequent surgical stress constitutes one of the key factors that suppresses immunity and affects the prognosis of GI patients [166,167]. Although surgical insults increasing the risk of tumor recurrence have been known for decades, the underlying mechanisms of treatment failure are still poorly understood. Neutrophils, as frontline responders after surgical stress, may play a pivotal role in inflammation and cancer progression [168,169]. Specifically, neutrophils may induce surgical stress through the formation of NETs. In vitro studies observed the adhesion of cancer cells to NET structures, indicating that part of the adverse effects of surgical stress might be due to NET formation [170]. The relevant statistics of clinical studies also show that the increase in the production of NETs is related to postoperative complications, such as prolonged hospitalization and increased mortality [171]. 

As for the mechanism of NETs affecting surgical stress, NETs may interact with the coagulation system, leading to poor surgical outcomes. Hongji Zhang is dedicated to researching NETs after surgical stress. His group found tumor capture and growth can be promoted by the influx of neutrophils after surgery [172]. Surgical stress may facilitate tumor metastasis through immunothrombosis, which results from the interaction between NETs and the coagulation system [173,174]. Research by Seth et al. was one of the first experimental studies to illustrate that surgery promoted tumor metastasis via a coagulation-dependent mechanism [175]. Hongji Zhang also demonstrated that NETs could activate platelets through platelet TLR4 in vivo [142]. With their ability to activate platelets to cause thrombus, NETs participate in various pathways leading to thrombus as intermediate substances. Zhang’s subsequent study showed that the platelet TLR4–ERK5 axis can facilitate the capturing of circulating tumor cells and distant metastasis after surgical stress and this process is mediated by NETs [176]. Their recent research found that HMGB1-activated platelets through TLR4 to influence NET formation [103]. HMGB1 not only binds to platelets through TLR4 but also activates TLR9-dependent pathways in tumor cells to produce the intermediate NETs [177,178]. Increased formation of NETs can accelerate cancer cells adhesion, proliferation, invasion, and migration after surgical stress. 

Since surgery remains the most suitable and only promising treatment for patients with metastatic disease, what we can do is to better understand the underlying mechanisms associated with post-surgical tumor recurrence and prevent recurrence and worsening of the situation. According to the above, NETs are the key factors triggering surgical stress and postoperative recurrence. Therefore, utilizing NETs as a therapeutic target to treat surgical stress may be an effective treatment option. Using DNase to inhibit NETs is a potential approach for further clinical applications. In a mouse model of surgical stress and liver metastasis, DNase I inhibition of NET formation has been shown to reduce the development of postoperative metastases [103]. Recombinant human DNase (rhDNase) has been previously used in a placebo-controlled, randomized trial in patients with autoimmune systemic lupus erythematosus disease [179]. rhDNase administration was well tolerated with no significant adverse events. This result suggests that DNAse may decrease the risk of recurrence in patients who have undergone resection with metastatic cancer.

## 6. The Detection and Quantification of NETs

From the above studies, we know that NETs play a non-negligible role in gastrointestinal tumors, thereby, the methods of NET detection and quantification are particularly important. At present, the most commonly used detection methods for NETs are use fluorescence microscopy to identify specific markers: citrullinated histone H3 (citH3) or MPO, NE, and DNA complexes (MPO-DNA, NE-DNA) [180]. However, these methods inevitably have limitations. For example, the results can be biased by observers and the detection time is generally long. With the deepening of related research, new methods are also currently being developed for clinical purposes, including ELISA-based assays [181], automated high-throughput live detection of nuclear changes [182], microfluidics [183], and integrated inertial-impedance cytometry [184]. Among them, microfluidic analysis tools can more accurately measure the number of NETs in the sample. They can mechanically capture NETs from blood samples and measure the number of intact NETs without measurement bias through the presence of degraded NETs in the sample [171]. In addition, Rebecca et al. reported a method that can preliminarily judge whether NETs are generated. The method utilizes pulse-shaped analysis (PulSA) to distinguish resting neutrophils from those with decondensed DNA, which is a prerequisite for NET formation [185]. Therefore, we can rule out the generation of NETs in patients and their associated pathological analysis by detecting the absence of neutrophils with decondensed DNA.

Noteworthily, imaging flow cytometry (IFC) as an emerging technology goes beyond the qualitative limitations of traditional flow cytometry [186,187] to enable high-throughput analysis. IFC combines the statistical robustness and the cellular detection of traditional flow cytometry with the analytical power of fluorescence microscopy, which has the added advantage of being able to assess early processes leading up to the extrusion of the DNA-scaffolded strands. With the quantitative capabilities of IFC, Emilia explored a novel IFC-based method for the specific quantification and detection of H4cit3. H4cit3 is a recognized marker of NETosis. The formation level of NETs can be obtained indirectly by quantitatively measuring H4cit3. Although this method is fast and with higher throughput, it is limited to the detection and quantification of H4cit3-mediated nuclear events [188]. Therefore, a novel IFC method was developed to measure NETs in vitro and whole blood samples. The method can distinguish NETosis from other types of cell death allowing accurate identification of NETs and NET precursors [189]. 

The recent development of computational methods for quantifying NETs has greatly improved the ability to study NETs. These methods range from using ImageJ to user-friendly applications and even more sophisticated machine learning methods. One of the computational methods for quantifying NETs, DNA Area and NETosis Analysis (DANA), is a novel ImageJ/Java-based program [190]. This computational approach offers an easy semi-automated method to quantify NET-like structures and DNA regions. Compared to using 3-dimensional confocal scanning laser microscopy (3D-CSLM) [191], DANA can sufficiently exclude overlapping cells and fragments that may be identified as false positives. Using different fluorescent markers to quantify DNA decondensation and the frequency of NET-like structures is a reliable, fast, simple, high-throughput, and cost-effective. This DNA staining method was further optimized and validated by Sarthak using the IncuCyte ZOOM imaging platform and the membrane-permeability properties of two DNA dyes. In this way, this optimized method can automatically quantify the percentage of neutrophils experiencing NETosis [182]. In addition, with DNA-staining dyes, the modern CNN model, a fully automated image-processing method for real-time imaging-based detection and quantification of NETs confirms that the inhibitory effect of GW 311616A (neutrophil elastase inhibitor) on the release of NETs [192]. Furthermore, the latest study has described an imaging and computational algorithm using a high-content screening (HCS)-cellomics platform. This platform employed membrane-permeable and impermeable DNA dyes in situ to identify NET-forming cells [193]. These scenarios illustrate the effectiveness of the combination of advanced computational methods and flexible staining methods for the quantification and study of NETs. (Figure 2, Table 1).

## 7. NETs in COVID-19 and NET-Associated Inhibitor Drugs

Coronavirus disease 2019 (COVID-19), has induced a pandemic with more than 620 million confirmed cases and 6.5 million deaths worldwide by October 2022. Due to the global pandemic of COVID-19, there has been considerable scientific interest in determining the interdependence of COVID-19 in cancer progression, as well as investigating the increasing risk of potential and life-threatening consequences of COVID-19 infection with the underlying medical conditions of cancer. A recent systematic review of 52 studies on COVID-19 and cancer showed that cancer patients have a high probability of dying from severe acute respiratory syndrome coronavirus 2 (SARS-CoV-2) [195]. COVID-19 was initially considered merely a respiratory disease. However, the digestive system may be involved, with an incidence ranging from 3% to 79% [196,197]. A review has demonstrated that any part of the digestive system can be affected by the SARS-CoV-2 virus. Gastrointestinal symptoms are also commonly seen in patients with COVID-19 in addition to the well-known respiratory symptoms [198]. Furthermore, a meta-analysis has demonstrated that lung cancer and CRC patients are more likely to be infected with SARS-CoV-2 than other cancer types, including breast, esophageal, bladder, pancreatic, and cervical cancers [199]. In addition to cancers of the digestive system that increase susceptibility to SARS-CoV-2, SARS-CoV-2 can also cause lesions in the digestive system such as liver damage and liver complications [200], as well as pancreatic damage. SARS-CoV-2 can directly damage the pancreas via ACE2 receptors and indirectly through locoregional vasculitis and thrombosis [201]. Emanuele Sinagra has carried out a full narrative review of COVID-19 and the pancreas to establish possible underlying mechanisms and scientific evidence supporting the association between COVID-19 and pancreatic disease [202]. The expression of angiotensin I-converting enzyme 2 (ACE2) which has been proven to be a cell receptor for SARS-CoV-2 is proposed as a key mediator of the virus in the body. Interestingly, a recent study found that ACE2 protein is abundantly expressed in the glandular cells of duodenal, rectal, and gastric epithelia, supporting SARS-CoV-2 entry into the host cells. It provided evidence for gastrointestinal infection of SARS-CoV-2 [203]. Moreover, ACE2 is expressed at a higher level in patients with GI cancer than in normal healthy individuals. For instance, the expression of ACE2 is found to be higher in CRC tissues than in matched normal tissues [204]. Due to the higher expression of ACE2 in lung metastases from CRC than in normal lungs, CRC patients are more susceptible to SARS-CoV-2 infection [203]. In summary, COVID-19 can further exacerbate and worsen cancers of the digestive system, and, in turn, cancers of the digestive system can promote COVID-19 infection.

It is a well-established concept that neutrophilia is as an indicator of severe respiratory symptoms and adverse outcomes in COVID-19 patients [205,206,207]. It has been demonstrated that neutrophil activation and degranulation are the most activated processes in SARS infection using gene ontology (GO) analysis based on the GSE1739 dataset in a transcriptomic profile study [208]. Similarly, neutrophils in SARS-CoV-2-infected patients are also heavily activated and produce a large number of immune-inflammatory products. In the highly inflammatory environment of patients infected with SARS-CoV-2, the presence of high concentrations of inflammatory mediators in the alveolar space leads to an influx of neutrophils and their subsequent activation, which promotes the release of reactive oxygen species (ROS), proteases, cytokines, and NETs. NETs also play a role in COVID-19 infection. A study has found the presence of NETs in the lungs of post-mortem COVID-19 patients [209] while another report demonstrated the role of NETs in immune thrombosis in COVID-19 patients [10]. Research has found that cancer patients may be more susceptible to SARS-CoV-2 infection due to immune suppression. SARS-CoV-2, like other viruses, can directly stimulate NET formation through unknown mechanisms. However, a recent study found that ROS formed during SARS-CoV-2 infection directly activated NETs, forming an inflammatory cascade in a vicious circle [210]. The release of NETs by neutrophils may be related to the ACE2/TMPRSS2 pathway shown by another study [194]. In addition, the ACE2/TMPRSS2 pathway is also significant in SARS-CoV-2 entry. This may explain why the gastrointestinal tract with increased expression of ACE2 is susceptible to COVID-19. We speculate that the release of NETs based on the ACE2/TMPRSS2 pathway may play an important role in the deterioration of GI cancer patients infected with COVID-19, which needs further research to demonstrate.

NETs have also been described as potential biomarkers of COVID-19 prognosis [9,10,211,212]. Almost all studies have shown that NETs can be induced in patients with COVID-19. More importantly, the highest levels of NETs were observed in critically ill patients, presumably correlating the level of NETs with the severity of the disease [9,213]. A study of COVID-19 patients with non-severe (NS), severe (S), and acute phase (PAP) disease found that the diagnostic power of NE and MPO (markers of NETs) as determined by the area under the subject operating curve (AUROC) was greater than 0.94, especially the diagnostic power of MPO in both severe (S) and acute phase (PAP) groups was shown to be 100%. This study shows the validity and accuracy of NETs as biomarkers for detection and diagnosis. This also illustrates the need to monitor these markers in all COVID-19 PAP individuals [214].

Considering the crucial role of NET in disease progression caused by SARS-CoV-2, it is extremely urgent to focus research on the detailed mechanism of action of NETs affecting the disease and the development of drugs targeting specific steps in the mechanism of action. A review summarized a series of targeted agents for NETs in COVID-19 patients, including metformin, vitamin D, aspirin, N-acetylcysteine, etc. [215]. Clinical trials have shown good results with these NET-targeted drugs in the treatment of patients with COVID-19 [216]. In Table 2, we list several clinical trials using drugs related to the inhibition of NETs for the treatment of GI tumors from the website https://clinicaltrials.gov/ accessed on 6 November 2022. There are a large number of registered clinical trials including a drug related to the inhibition of NETs, though few of them are currently completed. We will need to wait for the final results of these clinical trials to confirm whether these drugs that inhibit NETs can be effectively used for the treatment of gastrointestinal tract tumors, although preclinical data suggest that these drugs are promising targets for the treatment of GI tumors.

Discoveries about the impact of NETs on gastrointestinal tumors inspire and support the idea of potential therapeutic approaches that block NETs to effectively control tumor progression and metastasis. Among these emerging therapies, NET inhibitors in combination with immune checkpoint inhibitors for targeted treatment of gastrointestinal tumors may be a new therapeutic approach. It has been shown that DNase I can inhibit neutrophil extracellular traps, leading to a reversal of anti-PD-1 blockade resistance by increasing CD8+ T cell infiltration and cytotoxicity [217]. In addition, the antitumor efficiency of PD-1 inhibition could also be promoted by liraglutide that reduced NETs in lung and liver cancers [218]. Higher Neutrophil-to-Lymphocyte Ratio (NLR) is a reliable prognostic indicator for shorter overall survival in common metastatic cancers, and NLR is often directly associated with NETs. A study compared the efficacy of the currently marketed drugs tadalafil, isotretinoin, colchicine, and omega-3, and concluded that reducing NLR, or in other words, inhibiting NETs can slow down cancer growth [219]. Furthermore, with further research related to herbal medicine in the treatment of GI cancers, it is surprising to find that Huang Qin Tang could inhibit the initiation of colitis-associated carcinogenesis by controlling PAD4-dependent NETs. Thus, the herbal medicine Huang Qin Tang, can be used for the progressive prevention and treatment of GI cancers [220].

However, the current studies have some limitations. First, before putting into clinical trials, research on anti-NETs therapy mainly uses xenograft mouse models for experiments, which cannot realistically simulate the complex tumor microenvironment in tumor patients, so the factors considered may be limited [221]. Second, some drugs and therapies do not directly inhibit NETs but regulate the production of NETs through their upstream signaling molecules or cells. Each person may contain far more than one type of NETs, and some of these NETs are favorable. Therefore, due to the limited targeting ability of some drugs, they cannot directly target the specific kind of NETs and affect other unexpected molecules, which may produce some off-target effects [221,222]. Based on this limitation, further research and understanding of different classes of NETs associated with different cancers at different times and further development of targeted drugs are relatively important. In addition, combination therapy using NET inhibitors combined with some other drugs may be able to alleviate the corresponding side effects. In addition, although biomarkers of NETs are helpful in the prognosis of tumor patients, there is still no uniform clinical definition of normal and pathological levels of biomarkers of NETs as they are still not established [222]. Last but not least, inhibition of NETs is effective in attenuating cancer growth and metastasis, but since NETs are intrinsically pathogen-scavenging, inhibiting the function of NETs can have unwanted effects on the immune system [221].

## 8. Conclusions

NETs released by neutrophils were initially thought to act as an innate immune response to various infectious and inflammatory stimuli to combat microbial invasion, but an increasing number of studies have identified a pathological role for NETs in tumorigenesis and metastasis. Quantitative analysis of NETs by multiple methods can help us diagnose and prognose gastrointestinal cancers. Based on the understanding of the mechanisms by which NETs act on tumors, various drugs have been used in combination with immunotherapy to reduce the production of NETs and thus increase the survival rate of cancer patients.

However, we still need further studies to elucidate the detailed potential mechanisms of the interaction between NETs and gastrointestinal cancer. With a clearer and more specific understanding of the mechanism of action, more precise and efficient drugs can be developed to target specific NETs at different stages and types of tumors in the future. The use of safe and effective methods to transport drugs related to NETs inhibitors for the treatment of gastrointestinal cancers may also be an emerging research hotspot in the future.

## Figures and Tables

**Figure 1 ijms-24-00334-f001:**
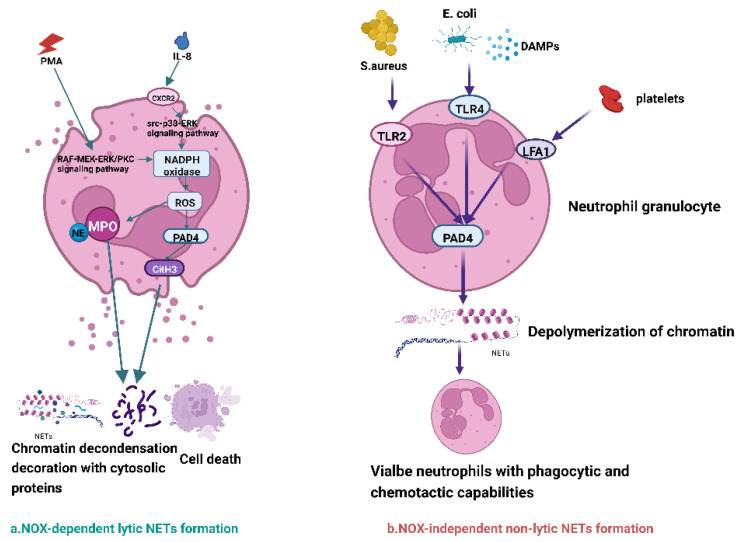
NETs are generated by neutrophils in two ways. The first way is through NADPH-oxidase (NOX)-dependent lytic NETs formation. When neutrophils are stimulated by factors such as phorbol myristate acetate (PMA), NADPH oxidase is activated through the RAF/MEK/ERK or the protein kinase C (PKC) signaling pathway, leading to the synthesis of reactive oxygen species (ROS) and the activation of protein arginine deaminase 4 (PAD4). PAD4 induces the formation of citrullinated histone H3 (CitH3) in the nucleus, which in turn promotes the depolymerization of chromatin and its release by exocytosis to form NETs [35]. IL-8 is an important cytokine that is released during tumor development. It has been reported that IL-8 interacts with CXC chemokine receptor 2 (CXCR2) on neutrophils, thereby triggering the formation of lytic NETs by activating the src–p38–ERK signaling pathway [36]. It is an oxidation-dependent pathway. In addition, MPO translocates to the nucleus with the help of NE [38] and promotes further chromatin depolymerization [39]. The nuclear envelope is decomposed, and the depolymerized chromatin gets released into the cytoplasm, modified by granules and cytoplasmic proteins, and then liberated from the cell. The second way is through NADPH-oxidase (NOX)-independent non-lytic NETs formation. The formation of such NETs is related to calcium influx and the generation of mitochondrial ROS independent of NOX activation [40]. Under the stimulation of *Staphylococcus aureus s* (*S. aureus*), *Escherichia coli* (*E. coli*), damage-associated molecular patterns (DAMPs), platelets, and Toll-like receptor 2/4 (TLR2/4), lymphocyte function-associated antigen 1 (LFA1) of neutrophils bind to these allergens, promoting the non-lytic NETs formation. This formation is also accompanied by chromatin de-concentration, followed by protein-modified chromatin expulsion through vesicles without disrupting the plasma membrane. The integrity of the nuclear and cellular membranes of neutrophils is not compromised under this NETs formation pathway. After releasing NETs, neutrophils are still alive and possess phagocytic and chemotactic capabilities [41]. This figure has been created with https://app.biorender.com (accessed on 28 November 2022).

**Figure 2 ijms-24-00334-f002:**
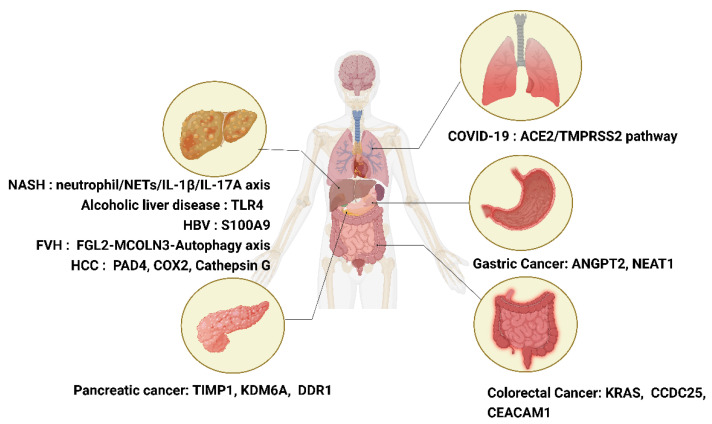
Specific targets of NETs in various diseases. This figure has been created with https://app.biorender.com (accessed 28 November 2022).

**Table 1 ijms-24-00334-t001:** Specific targets of NETs in various diseases and related mechanisms.

Specific Disease	Specific Targets	Mechanism
GC	ANGPT2	NETs upregulate the expression of ANGPT2 in human umbilical vein endothelial cells, then ANGPT2 activates TIE2-expressing monocytes/macrophages (TEM), which promote angiogenesis, tumor formation, metastasis, and immunosuppression [79].
	NEAT1	NETs can promote the invasion of GC cells by upregulating NEAT1 expression [84].
CRC	KRAS	Exosomes may transfer mutant KRAS to recipient cells and trigger increases in IL-8 production, neutrophil recruitment, and formation of NETs, eventually leading to the deterioration of CRC [57].
	CCDC25	NET-DNA binding to CCDC25 triggers the ILK–β-parvin–RAC1–CDC42 cascade reaction, thereby enhancing the development of cancer metastasis [105].
	CEACAM1	CEACAM1 on NETs can promote colon cancer cell adhesion, migration, and metastasis [106].
NASH	Neutrophil/NETs/IL-1β/IL-17A axis	NETs enriched with IL-1β and IL-17A participate in the hepatic inflammatory process of patients with NASH [119].
Alcohol liver disease	TLR4	Formation of NETs following LPS stimulation of TLR4 upon chronic alcohol use leads to increased alcoholic steatosis and subsequent HCC [123].
HBV	S100A9	The activation of RAGE/TLR4–ROS signaling by HBV-induced S100A9 results in abundant NET formation, which subsequently facilitates the growth and metastasis of HCC cells [125].
FVH	FGL2-MCOLN3-Autophagy axis	FGL2 directly interacts with mucolipin 3, which regulated calcium influx and initiates autophagy, leading to the formation of NETs [194].
HCC	PAD4	Knocking out the PAD4 gene or inhibiting the formation of NETs with drugs, significantly reduces the number and size of cancer cells on the liver surface [118].
	COX2	NETs induce the inflammatory response of HCC cells by up-regulating COX2 expression, thereby activating the TLR4/9 signaling pathway, and promoting the metastasis ability of liver cancer cells [127].
	Cathepsin G	NET-associated cathepsin G component enhances HCC metastasis [129].
PDAC	TIMP1	TIMP1 triggers NET formation in primary human neutrophils through mechanisms dependent on the interaction of TIMP1 with its receptor CD63 and subsequent ERK signaling [148].
	KDM6A	KDM6A loss in pancreatic cancer cells alters the immune microenvironment by increasing CXCL1 secretion and enhancing NET formation [151].
	DDR1	Collagen-induced DDR1 activation in cancer cells is a major stimulus for CXCL5 production, subsequently resulting in the recruitment of TANs, the formation of NETs, and eventually, cancer cell invasion and metastasis [158].
COVID-19	ACE2/TMPRSS2 pathway	SARS-CoV-2 directly stimulates neutrophils to release NETs, which is dependent on ACE2 and the serine protease activity axis and effective viral replication [194].

**Table 2 ijms-24-00334-t002:** Ongoing clinical trials in GI cancer patients with drugs that target the process of neutrophil extracellular trap (NETs) release.

NETs Inhibitors	Specific Cancer	NTC	Status	Target
Metformin	Colon Cancer	NCT03359681	Completed	HMGB1
		NCT01440127	Terminated	
		NCT02614339	Unknown	
		NCT01816659	Terminated	
		NCT01632020	Terminated	
		NCT04947020	Recruiting	
	Pancreatic Cancer	NCT01210911	Completed	
		NCT01167738	Terminated	
		NCT01954732	Withdrawn	
Fostamatinib	Colorectal Cancer and Liver Cancer	NCT00923481	Completed	NET release
Disulfiram	Pancreatic Cancer	NCT02671890	Recruiting	Gasdermin D
	Metastatic Pancreatic Cancer	NCT03714555	Completed	
Cholecalciferol	Colon Cancer and Rectal Cancer	NCT02603757	Completed	ROS
		NCT01198548	Terminated	
		NCT01403103	Withdrawn	
	Colorectal Cancer	NCT00470353	Terminated	
N-acetylcysteine	Gastrointestinal Tumor	NCT00196885	Completed	
	Liver Cancer	NCT01394497	Completed	
Vitamin D	Gastrointestinal Tumor	NCT05552729	Not yet recruiting	
		NCT00585637	Completed	
	Rectal Cancer	NCT04857203	Recruiting	
	Colorectal Cancer	NCT04868227	Completed	
		NCT01074216	Completed	
		NCT01150877	Completed	
	Liver Cancer	NCT02461979	Recruiting	
	Pancreas Cancer	NCT03472833	Terminated	
	Hepatocellular Carcinoma	NCT01575717	Unknown	
		NCT02779465	Not yet recruiting	
Aspirin	Gastrointestinal Tumor	NCT04081831	Completed	MAPK and NF-κB
	Colon Cancer	NCT02467582	Active, not recruiting	
	Colorectal Cancer Liver Metastases	NCT03326791	Recruiting	
	Gastric Cancer	NCT04214990	Recruiting	
	Colorectal Cancer	NCT02647099	Recruiting	
		NCT03579732	Completed	
		NCT00565708	Active, not recruiting	
Cyclosporine A	Solid Neoplasm	NCT02188264	Active, not recruiting	Calcineurin/Nuclear Factor of Activated T Cells
	Colorectal Cancer	NCT00003950	Completed	
		NCT00389870	Completed	
Colchicine	Liver Cancer	NCT01935700	Completed	Neutrophil
		NCT04264260	Unknown	
Hydroxychloroquine	Gastrointestinal Tumor	NCT05221320	Recruiting	
		NCT04214418	Recruiting	
	Colorectal Cancer	NCT01006369	Completed	
		NCT03215264	Completed	
	Pancreatic Cancer	NCT01273805	Completed	
		NCT01494155	Active, not recruiting	
		NCT05083780	Recruiting	
NSIADs	Gastric Cancer	NCT00172627	Unknown	Prostaglandins (PGs)
Etodolac	Colorectal Cancer	NCT03919461	Recruiting	
	Pancreatic Cancer	NCT03838029	Recruiting	
Celecoxi	Pancreatic Cancer	NCT00176813	Completed	
	Colorectal Cancer	NCT00582660	Completed	
		NCT00466505	Completed	
Heparin	Gastrointestinal Tumor	NCT02444572	Completed	Circulating Histone and NF-κB
	Liver Cancer	NCT00827554	Completed	
	Colorectal Cancer	NCT01589146	Unknown	
	Gastrointestinal Tumor	NCT05041335	Not yet recruiting	
	Pancreatic Cancer	NCT01945879	Completed	
	Gastric Cancer	NCT00718354	Completed	

## Data Availability

Data sharing not applicable to this article as no datasets were generated or analyzed during this study.

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
