# Peer review of "Emerging Role of Neutrophil Extracellular Traps in Gastrointestinal Tumors: A Narrative Review"

_ijms, 2022, doi:10.3390/ijms24010334_

Round 1

Reviewer 1 Report

The manuscript of “Neutrophil Extracellular Traps: Emerging Plays in Gastrointestinal Tumors” by Yujun Zhang and co-authors aims to review the involvement of Neutrophil Extracellular Traps (NETs) in the gastrointestinal oncologic microenvironment, proliferation and metastasis of gastrointestinal tumors (namely gastric, colorectal, pancreatic, and liver cancers), tumor-related thrombosis, and surgical stress. The authors also analyzed the prognostic value of NETs as a specific marker of coronavirus disease 2019 and the possibility of the use of NETs inhibitors for the treatment of COVID-19 and digestive tract tumor diseases. The topic of the manuscript is highly relevant and timely in view of recent the statistics on the incidence of gastric cancers in the world. The review covers a large body of literature and makes a significant contribution to the systematization of knowledge about the role of neutrophils and the NET production in the comprehensive treatment of gastrointestinal cancers. The manuscript is very interesting, easy to read and understand, written at a high professional level. The authors cited a large number of research papers, a significant portion of which have been published over the last five years. The manuscript is well structured; all the conclusions are supported by recent data.

Minor comments.

Figure 1: Some stylistic and punctuation errors in the title and legend need to be corrected. It would be better to add a more specific name for the two mechanisms of NETs formation and provide corresponding literary references in the legend.

Figure 2: The title should be placed under the Figure. It would be better to add a legend with a more detailed description and a corresponding literary reference to each of the mechanisms of activation of signaling pathways given.

Author Response

Dear reviewer:

Thank you for your comments concerning our manuscript entitled “Neutrophil Extracellular Traps: Emerging Plays in Gastrointestinal Tumors” (IJMS-2097847). All your suggestions are very important to us, both for composing the manuscript and for our further research. We have studied the comments carefully and have made corrections which we hope meet with approval. Based on your advice, we amended the relevant section in the manuscript. All your questions are answered below.

  1. Figure 1: Some stylistic and punctuation errors in the title and legend need to be corrected. It would be better to add a more specific name for the two mechanisms of NETs formation and provide corresponding literary references in the legend.

Response: We are grateful for the suggestion, and we have modified Figure 1 accordingly. The corresponding literary references in the legend are provided in Figure 1. What’s more, we have access to relevant literature to describe in more detail the two ways in which NETs are formed(Line 112-135). We provide a more specific name for the two mechanisms of NETs formations. The two types of mechanism are NADPH-oxidase (NOX)-dependent lytic NETs formation and NADPH-oxidase (NOX)-independent non-lytic NETs formation.

  1. Figure 2: The title should be placed under the Figure. It would be better to add a legend with a more detailed description and a corresponding literary reference to each of the mechanisms of activation of signaling pathways given.

Response: We apologize for this problem in the original manuscript. We make a simple modification to Figure 2 and place it below the title. To introduce the mechanisms of activation of signaling pathways more clearly and in detail, we have briefly introduced the mode of action of the target molecules in Table 1 and cited the relevant references in the table.

Thank you very much for all your help and looking forward to hearing from you soon.

Best regards

Yours sincerely

Zhen Zong

Reviewer 2 Report

The topic of the manuscript is the literature review on the role of the Neutrophil Extracellular Traps in gastrointestinal tumours.

The abstract and the main text of the article are informative. The review is interestingly written, however, some following points must be clarified/corrected for the further processing of this manuscript.

Merits-related comments:

1.       The title should be supplemented by the phrase “narrative review”.

2.       Also, keywords should be enriched with the proper MeSH terms.

3.       Clarify the purpose of the review in a separate paragraph.

4.       Although the manuscript does not constitute a systematic review, it would be good to add a paragraph on the search and selection of the references included in the review, using part of the PRISMA guidelines.

5.       Please cite current “Cancer Statistics, 2020” instead of older versions of 2016 and 2019.

6.       Moreover, it is suggested to remove section about COVID-19 (it seems to be not consistent with the theme of the review).

7.       In the introductory sections, it would be useful to add the paragraph about the role of NETs in other gastroenterological diseases, e.g. inflammatory bowel diseases (the recommended recent references: 10.3390/life11121409, 10.3390/life11090943, 10.1016/j.jcmgh.2021.03.002, 10.1186/s13099-022-00497-x etc.).

Technical comments:

1.       The manuscript requires editorial editing, e. g. typing errors, missing spaces before references.

2.       Are all figures created by the Authors of the manuscript? The annotation in the captions would be useful.

3.       References should be described as follows:
1. Author 1, A.B.; Author 2, C.D. Title of the article. 
Abbreviated Journal Name YearVolume, page range.

Author Response

Dear reviewers:

We are very grateful for your comments concerning our manuscript entitled “Neutrophil Extracellular Traps: Emerging Plays in Gastrointestinal Tumors” (IJMS-2097847). The comments are valuable and very helpful. We have studied the comments carefully and have made corrections which we hope meet with approval. Based on your advice, we amended the relevant section in the manuscript. All your questions are answered below.

Merits-related comments:

  1. The title should be supplemented by the phrase narrative review.

Response: We are grateful for the suggestion, and we renamed the article title to “Neutrophil Extracellular Traps are Playing An Emerging Role in Gastrointestinal Tumors: a narrative review”

  1. 2. Also, keywords should be enriched with the proper MeSH terms.

Response: We are grateful for the suggestion, and to be more clear and in accordance with your concerns, we have revised the keywords with the proper MeSH terms. The revised keywords are neutrophil extracellular traps; gastrointestinal neoplasms; molecular targeted therapy; biomarkers; quantitation.

  1. Clarify the purpose of the review in a separate paragraph.

Response: Thank you for your comments. Based on your advice, we have clarified the purpose of the review in a separate paragraph (Line 71-81).

  1. Although the manuscript does not constitute a systematic review, it would be good to add a paragraph on the search and selection of the references included in the review, using part of the PRISMA guidelines.

Response: Thank you very much for your comments. Searches were conducted using the database Medline (via PubMed). Search terms included “neutrophil Extracellular Traps”, “NETs”, “gastrointestinal cancer”, “gastrointestinal carcinoma”, “gastrointestinal neoplasms”, “gastric cancer”, “gastric carcinoma”, “gastric neoplasms”, “colorectal cancer”, “colorectal carcinoma”, “colorectal neoplasms”, “liver cancer”, “liver carcinoma”, “liver neoplasms”, “pancreatic cancer”, “pancreatic carcinoma”, “pancreatic neoplasms”, “therapy”, “biomarkers”, “ quantitation”, and “detection” in diverse combinations. Search terms were connected using Boolean operators AND and OR to capture all relevant articles.

If you think it is necessary to add a paragraph on the search and selection of the references included in the review, we will add it later.

  1. Please cite current “Cancer Statistics, 2020” instead of older versions of 2016 and 2019.

Response: Thank you very much for your comments. We removed the citations “Cancer Statistics, 2016”, “Cancer Statistics, 2018”, “Cancer Statistics, 2019”. We only cite “Cancer Statistics, 2020” to analyze the current status of gastrointestinal cancers (Line 34-44).

  1. Moreover, it is suggested to remove section about COVID-19 (it seems to be not consistent with the theme of the review).

Response: We apologize for not highlighting the role of NETs in COVID-19 with gastrointestinal cancers when we introduced it, and for causing you this confusion. For this reason, we have revised the manuscript accordingly so that you can clearly understand their relevance (Line 762-765,825-831). However, if you feel the need to remove this section, we will remove it later.

First, we found a mutually reinforcing relationship between COVID-19 and gastrointestinal cancer. It was shown by researchers that any part of the digestive system can be affected by the SARS-CoV-2 virus. Furthermore, studies have shown that SARS-CoV-2 further exacerbates and worsens digestive cancers. These findings suggest that COVID-19 promotes further exacerbation of gastrointestinal cancer. In turn, digestive cancers promote SARS-CoV-2 infection. For instance, we found that colorectal cancer patients are more susceptible to SARS-CoV-2 infection.

Next, we hypothesized that the release of NETs based on the ACE2/TMPRSS2 pathway may play an important role in patients with gastrointestinal cancers infected with COVID-19 based on the relevant literature.

Then, we identified NET as a potential biomarker of COVID-19 prognosis, and we found some clinical trials have been conducted to demonstrate the significant role of NET-targeted drugs in the treatment of COVID-19. Surprisingly, we found these drugs also have a significant role of these drugs in the treatment of gastrointestinal cancers in clinical trials. As a result, we speculated that these drugs have a dual role in the treatment of gastrointestinal cancer patients suffering from COVID-19. This also partly echoes the aforementioned link between COVID-19 and gastrointestinal cancer.

Finally, the mutual promotion relationship between COVID-19 and gastrointestinal cancer provides new insights into the pathogenesis of gastrointestinal cancer and shows the added value and special significance of gastrointestinal cancer treatment in COVID-19-infected patients. What’s more, this finding also warns us of the higher requirements for the prevention of gastrointestinal diseases in COVID-19-infected patients.

  1. In the introductory sections, it would be useful to add the paragraph about the role of NETs in other gastroenterological diseases, e.g. inflammatory bowel diseases (the recommended recent references: 10.3390/life11121409, 10.3390/life11090943, 10.1016/j.jcmgh.2021.03.002, 10.1186/s13099-022-00497-x etc.).

Response:  Thank you very much for your comments. After learning the articles that you have cited, we firstly cited them in “1 Introduction” in a yellow background (Line 61-65): “Inflammatory bowel disease (IBD)[10.3390/life11121409], ulcerative colitis (UC) and Crohn's disease (CD)[10.3390/life11090943] are characterized by chronic unresolved intestinal mucosal inflammation involving innate and adaptive immune responses. Neutrophils are key players in the inflammatory milieu of the intestinal mucosa in IBD, enhancing and perpetuating intestinal inflammation by forming NETs[10.1016/j.jcmgh.2021.03.002, 10.1186/s13099-022-00497-x ]”.

Technical comments:

1.The manuscript requires editorial editing, e. g. typing errors, missing spaces before references.

Response: We apologize for this problem in the original manuscript, and we have revised the errors in our manuscript marked with red font.

2.Are all figures created by the Authors of the manuscript? The annotation in the captions would be useful.

Response: Thank you very much for your comments.All figures are created by the authors of the manuscript from https://app.biorender.com

3.References should be described as follows:

  1. Author 1, A.B.; Author 2, C.D. Title of the article. Abbreviated Journal Name Year, Volume, page range.

Response: We apologize for this problem in the original manuscript, and we have revised the format of references according to your opinions and the standards of the journal.

We would love to thank you for allowing us to resubmit a revised copy of the manuscript and we highly appreciate your time and consideration.

Best regards

Yours sincerely

Zhen Zong

Round 2

Reviewer 2 Report

The Authors have taken into account all the suggestions of the Reviewers and improved the manuscript considerably. I have no further comments.